# ADAPTIVE GRADIENT METHODS WITH LOCAL GUARANTEES

## ABSTRACT

Adaptive gradient methods are the method of choice for optimization in machine learning and used to train the largest deep models. In this paper we study the problem of learning a local preconditioner, that can change as the data is changing along the optimization trajectory. We propose an adaptive gradient method that has provable adaptive regret guarantees vs. the best local preconditioner. To derive this guarantee, we prove a new adaptive regret bound in online learning that improves upon previous adaptive online learning methods.

We demonstrate the practical value of our algorithm for learning rate adaptation in both online and offline settings. For the online experiments, we show that our method is robust to unforeseen distribution shifts during training and consistently outperforms popular off-the-shelf learning rate schedulers. For the offline experiments in both vision and language domains, we demonstrate our method's robustness and its ability to select the optimal learning rate on-the-fly and achieve comparable task performance as well-tuned learning rate schedulers, albeit with less total computation resources.

## 1 INTRODUCTION

Adaptive gradient methods have revolutionized optimization for machine learning and are routinely used for training deep neural networks. These algorithms are stochastic gradient based methods, that also incorporate a changing data-dependent preconditioner (multi-dimensional generalization of learning rate). Their empirical success is accompanied with provable guarantees: in any optimization trajectory with given gradients, the adapting preconditioner is comparable to the best in hindsight, in terms of rate of convergence to local optimality.

Their success has been a source of intense investigations over the past decade, since their introduction, with literature spanning thousands of publications, some highlights are surveyed below. The common intuitive understanding of their success is their ability to change the preconditioner, or learning rate matrix, per coordinate and on the fly. A methodological way of changing the learning rate allows treating important coordinates differently as opposed to commonly appearing features of the data, and thus achieve faster convergence.

In this paper we investigate whether a more refined goal can be obtained: namely, can we adapt the learning rate per coordinate, and also in short time intervals? The intuition guiding this question is the rising popularity in "exotic learning rate schedules" for training deep neural networks. The hope is that an adaptive learning rate algorithm can automatically tune its preconditioner, on a per-coordinate and per-time basis, such to guarantee optimal behavior even locally.

To pursue this goal, we use and improve upon techniques from the literature on adaptive regret in online learning to create a provable method that is capable of attaining optimal regret in any sub-interval of the optimization trajectory. We then test the resulting method and compare it to learning a learning rate schedule from scratch. Experiments conducted validate that our algorithm can improve accuracy and robustness upon existing algorithms for online tasks, and for offline tasks it saves overall computational resources for hyperparameter optimization.

## 1.1 STATEMENT OF OUR RESULTS

The (stochastic/sub)-gradient descent algorithm is given by the following iterative update rule:

$$x_{\tau+1} = x_\tau - \eta_\tau \nabla_\tau.$$

If $\eta_\tau$ is a matrix, it is usually called a preconditioner. A notable example for a preconditioner is when $\eta_\tau$ is equal to the inverse Hessian (or second differential), which gives Newton's method. Let $\nabla_1, ..., \nabla_T$ be the gradients observed in an optimization trajectory, the Adagrad algorithm (and subsequent adaptive gradient methods, notably Adam) achieves the following regret guarantee for online convex optimization (OCO):

$$\tilde{O}(\sqrt{\min_{H \in \mathcal{H}} \sum_{\tau=1}^{T} \|\nabla_\tau\|_H^{*2}}),$$

where $\mathcal{H}$ is a family of matrix norms, most commonly those with a bounded trace. In this paper we propose a new algorithm SAMUEL, which improves upon this guarantee in terms of the local performance over any sub-interval of the optimization trajectory. For any sub-interval $I = [s, t]$, the regret over $I$ can be bounded by

$$\tilde{O}(\sqrt{\min_{H \in \mathcal{H}} \sum_{\tau=s}^{t} \|\nabla_\tau\|_H^{*2}}),$$

which also implies a new regret bound over $[1, T]$:

$$\tilde{O}\left( \min_k \min_{H_1,...,H_k \in \mathcal{H}} \sum_{j=1}^{k} \sqrt{\sum_{\tau \in I_j} \|\nabla_\tau\|_{H_j}^{*2}} \right)$$

This regret can be significantly lower than the regret of Adagrad, Adam and other global adaptive gradient methods that do not perform local optimization to the preconditioner. We spell out such a scenario in the next subsection.

Our main technical contribution is a variant of the multiplicative weight algorithm, that achieves full-matrix regret bound over any interval by automatically selecting the optimal local preconditioner. The difficulty in this new update method stems from the fact that the optimal multiplicative update parameter, to choose the best preconditioner, depends on future gradients and cannot be determined in advance. To overcome this difficulty, we run in parallel many instantiations of the update rule, and show that this can be done albeit increasing the number of base adaptive gradient methods by only a logarithmic factor. A comparison of our results in terms of adaptive regret is given in Table 1.

We conduct experiments in optimal learning rate scheduling to support our theoretical findings. We show that for an online vision classification task with distribution shifts unknown to the learning algorithm, our method achieves better accuracy than previous algorithms. For offline tasks, our method is able to achieve near-optimal performance robustly, with fewer overall computational resources in hyperparameter optimization.

## 1.2 WHEN DO LOCAL GUARANTEES HAVE AN ADVANTAGE?

Our algorithm provides near optimal adaptive regret bounds for any sub-interval $[s, t] \subset [1, T]$ simultaneously, giving more stable regret guarantee for a changing environment. In terms of classical regret bound over the whole interval $[1, T]$, our algorithm obtains the optimal bound of Adagrad up to a $O(\sqrt{\log T})$ factor.

Moreover, adaptive regret guarantees can drastically improve the loss over the entire interval. Consider the following example in one dimension. For $t \in [1, \frac{T}{2}]$ the loss function is $f_t(x) = (x + 1)^2$ and for the rest of time it is $f_t(x) = (x - 1)^2$. Running a standard online gradient descent method that is known to be optimal for strongly convex losses, i.e. with $\eta_t = \frac{1}{t}$, gives an $O(\log T)$ regret. However, the overall loss is $\Omega(T)$ because the best comparator in hindsight is $x = 0$ which has overall loss $T$. However, if we have adaptive regret guarantees, the overall loss on both $[1, \frac{T}{2}]$ and $[\frac{T}{2} + 1, T]$ are both $O(\log T)$, which is a dramatic $O(T)$ improvement in regret.

| Algorithm | Regret over $I = [s, t]$ |
|---|---|
| Hazan & Seshadhri (2007) | $\tilde{O}(\sqrt{T})$ |
| Daniely et al. (2015), Jun et al. (2017) | $\tilde{O}(\sqrt{|I|})$ |
| Cutkosky (2020) | $\tilde{O}(\sqrt{\sum_{\tau=s}^{t} \|\nabla_\tau\|^2})$ |
| SAMUEL (ours) | $\tilde{O}(\sqrt{\sum_{\tau=s}^{t} \|\nabla_\tau\|_H^{*2}})$ |

Table 1: Comparison of results. We evaluate the regret performance of the algorithms on any interval $I = [s, t]$. For the ease of presentation we hide secondary parameters. Our algorithm achieves the regret bound of Adagrad, which is known to be tight in general, but on any interval.

## 1.3 RELATED WORK

Our work lies in the intersection of two related areas: adaptive gradient methods for continuous optimization, and adaptive regret algorithms for regret minimization, surveyed below.

**Adaptive Gradient Methods.** Adaptive gradient methods and the Adagrad algorithm were proposed in (Duchi et al., 2011). Soon afterwards followed other popular algorithms, most notable amongst them are Adam (Kingma & Ba, 2014) and RMSprop (Tieleman & Hinton, 2012). Despite significant practical impact, their properties are still debated Wilson et al. (2017).

Numerous efforts were made to improve upon these adaptive gradient methods in terms of parallelization, memory consumption and computational efficiency of batch sizes, e.g. (Shazeer & Stern, 2018; Agarwal et al., 2019; Gupta et al., 2018; Chen et al., 2019). A survey of adaptive gradient methods appears in Goodfellow et al. (2016); Hazan (2019).

**Adaptive Regret Minimization in Online Convex Optimization.** The concept of competing with a changing comparator was pioneered in the work of (Herbster & Warmuth, 1998; Bousquet & Warmuth, 2003) on tracking the best expert. Motivated by computational considerations for convex optimization, the notion of adaptive regret was first introduced by Hazan & Seshadhri (2007), which generalizes regret by considering the regret of every interval. They also provided an algorithm Follow-The-Leading-History which attains $\tilde{O}(\sqrt{T})$ adaptive regret. Daniely et al. (2015) considered the worst regret performance among all intervals with the same length and obtain $O(\sqrt{|I| \log^2 T})$ interval-length dependent bounds, improved later by Jun et al. (2017) and Cutkosky (2020).

For other related work, some considered the dynamic regret of strongly adaptive methods Zhang et al. (2018; 2020). Zhang et al. (2019) considered smooth losses and proposes SACS which achieves an $O(\sum_{\tau=s}^{t} \ell_\tau(x_\tau) \log^2 T)$ regret bound.

**Learning Rate Schedules and Hyperparameter Optimization.** On top of adaptive gradient methods, a plethora of nonstandard learning rate schedules have been proposed. A commonly used one is the step learning rate schedule, which changes the learning rate at fixed time-points. A cosine annealing rate schedule was introduced by Loshchilov & Hutter (2016). Alternative learning rates were studied in Agarwal et al. (2021). Learning rate schedules which increase the learning rate over time were proposed in Li & Arora (2019). Learning the learning rate schedule itself was studied in Wu et al. (2018). Large-scale experimental evaluations (Choi et al., 2019; Schmidt et al., 2020; Nado et al., 2021) conclude that hyperparameter optimization over the learning rate schedules are essential to state-of-the-art performance.

## 2 SETTING AND PRELIMINARIES

**Online convex optimization.** Consider the problem of online convex optimization (see Hazan (2016) for a comprehensive treatment). At each round $\tau$, the learner outputs a point $x_\tau \in \mathcal{K}$ for some convex domain $\mathcal{K} \subset R^d$, then suffers a convex loss $\ell_\tau(x_\tau)$ which is chosen by the adversary. The learner also receives the sub-gradients $\nabla_\tau$ of $\ell_\tau()$ at $x_\tau$. The goal of the learner in OCO is to

---

**Algorithm 1** Strongly Adaptive regularization via MUltiplicative-wEights (SAMUEL )

---

Input: OCO algorithm $\boldsymbol{A}$, geometric interval set $S$, constant $Q = 4\log(dTD^2G^2)$.
Initialize: for each $I \in S$, $Q$ copies of OCO algorithm $\boldsymbol{A}_{I,q}$.
Set $\eta_{I,q} = \frac{1}{2GD2^q}$ for $q \in [1, Q]$.
Initialize $w_1(I, q) = \min\{1/2, \eta_{I,q}\}$ if $I = [1, s]$, and $w_1(I, q) = 0$ otherwise for each $I \in S$.
**for** $\tau = 1, \ldots, T$ **do**
    Let $x_\tau(I, q) = \boldsymbol{A}_I(\tau)$
    Let $W_\tau = \sum_{I \in S(\tau), q} w_\tau(I, q)$.
    Let $x_\tau = \sum_{I \in S(\tau), q} w_\tau(I, q) x_\tau(I, q) / W_\tau$.
    Predict $x_\tau$.
    Receive loss $\ell_\tau(x_\tau)$, define $r_\tau(I) = \ell_\tau(x_\tau) - \ell_\tau(x_\tau(I, q))$.
    For each $I = [s, t] \in S$, update $w_{\tau+1}(I, q)$ as follows,

$$w_{\tau+1}^{(I,q)} = \begin{cases} 0 & \tau + 1 \notin I \\ \min\{1/2, \eta_{I,q}\} & \tau + 1 = s \\ w_\tau(I, q)(1 + \eta_{I,q} r_\tau(I)) & \textbf{else} \end{cases}$$

**end for**

---

minimize regret, defined as

$$\text{Regret} = \sum_{\tau=1}^{T} \ell_\tau(x_\tau) - \min_{x \in \mathcal{K}} \sum_{\tau=1}^{T} \ell_\tau(x).$$

Henceforth we make the following basic assumptions for simplicity (these assumptions are known in the literature to be removable):

**Assumption 1.** *There exists $D, D_\infty > 1$ such that $\|x\|_2 \leq D$ and $\|x\|_\infty \leq D_\infty$ for any $x \in \mathcal{K}$.*

**Assumption 2.** *There exists $G > 1$ such that $\|\nabla_\tau\|_2 \leq G, \forall \tau \in [1, T]$.*

We make the notation of the norm $\|\nabla\|_H$, for any PSD matrix $H$ to be:

$$\|\nabla\|_H = \sqrt{\nabla^\top H \nabla}$$

And we define its dual norm to be $\|\nabla\|_H^* = \sqrt{\nabla^\top H^{-1} \nabla}$. In particular, we denote $\mathcal{H} = \{H | H \succeq 0, tr(H) \leq d\}$. We consider Adagrad from Duchi et al. (2011), which achieves the following regret if run on $I = [s, t]$:

$$\text{Regret}(I) = O\left( Dd^{\frac{1}{2}} \min_{H \in \mathcal{H}} \sqrt{\sum_{\tau=s}^{t} \nabla_\tau^\top H^{-1} \nabla_\tau} \right)$$

**The multiplicative weight method.** The multiplicative weight algorithm is a generic algorithmic methodology first used to achieve vanishing regret for the problem of prediction from expert advice Littlestone & Warmuth (1994). Various variants of this method are surveyed in Arora et al. (2012), that attain expert regret of $O(\sqrt{T \log(N)})$ for binary prediction with $N$ experts.

## 3 An Improved Adaptive Regret Algorithm

In this section, we describe the SAMUEL algorithm 1, which combines a novel variant of multiplicative weight as well as adaptive gradient methods to obtain stronger regret bounds in online learning and optimization.

The SAMUEL algorithm 1 guarantees that given any black-box OCO algorithm $\boldsymbol{A}$ as experts, achieves an

$$\tilde{O}\left( \sqrt{\min_{H \in \mathcal{H}} \sum_{\tau=s}^{t} \nabla_\tau^\top H^{-1} \nabla_\tau} \right)$$

regret bound (w.r.t. the experts) over any interval $J = [s, t]$ simultaneously. Next, by setting Adagrad as the black-box OCO algorithm $\boldsymbol{A}$, the above bound matches the regret of the best expert and holds w.r.t. any fixed comparator as a result, implying an optimal full-matrix adaptive regret bound.

Roughly speaking, Algorithm 1 first picks a subset $S$ of all sub-intervals and initiates an instance of the black-box OCO algorithm $\boldsymbol{A}$ on any interval $I \in S$ as an expert. The expert for interval $I$ is especially designed to achieve optimal regret over $I$ instead of $[1, T]$. To improve upon previous works and achieve the full-matrix regret bound, we make $O(\log T)$ duplicates of each expert with different decaying factors $\eta$, which is the main novel mechanism of our algorithm (notice that these duplicates share the same model therefore won't bump up computational cost). Then Algorithm 1 runs a multiplicative weight update on all active experts $\mathcal{A}_{I,q}$ denoting the expert over $I$ with the $q$-th decaying factor $\eta$ (if $\tau \in I$) according to the loss of their own predictions, normalized by the loss of the true output of the algorithm.

We follow Daniely et al. (2015) on the construction of $S$: without loss of generality, we assume $T = 2^k$ and define the geometric covering intervals following Daniely et al. (2015):

**Definition 1.** Define $S_i = \{[1, 2^i], [2^i + 1, 2^{i+1}], ..., [2^k - 2^i + 1, 2^k]\}$ for $0 \leq i \leq k$. Define $S = \cup_i S_i$ and $S(\tau) = \{I \in S | \tau \subset I\}$.

For $2^k < T < 2^{k+1}$, one can similarly define $S_i = \{[1, 2^i], [2^i + 1, 2^{i+1}], ..., [2^i \lfloor \frac{T-1}{2^i} \rfloor + 1, T]\}$, see Daniely et al. (2015). The intuition behind using $S$ is to reduce the $\Omega(T)$ computational cost of the naive method which constructs an expert for every subinterval of $[1, T]$. Henceforth at any time $\tau$ the number of 'active' intervals is only $O(\log(T))$, this guarantees that the running time and memory cost per round of SAMUEL is as fast as $O(\log(T))$. Decompose the total regret over an interval $J$ as $R_0(J) + R_1(J)$, where $R_0(J)$ is the regret of an expert $\boldsymbol{A}_J$ and $R_1(J)$ is the regret of the multiplicative weight algorithm 1. Our main theoretical result is the following:

**Theorem 2.** *Under assumptions 1 and 2, the regret $R_1(J)$ of the multiplicative weight part in Algorithm 1 satisfies that for any interval $J = [s, t]$,*

$$R_1(J) = O\left(D \log(T) \max\left\{G\sqrt{\log(T)}, d^{\frac{1}{2}}\sqrt{\min_{H \in \mathcal{H}} \sum_{\tau=s}^{t} \|\nabla_\tau\|_H^{*2}}\right\}\right)$$

**Remark 3.** We note that $q$ that $r_\tau(I, q)$ and $x_\tau(I, q)$ doesn't depend on $q$ for the same $I$, so we may write $r_\tau(I)$ and $x_\tau(I)$ instead for simplicity. We use convex combination in line 8 of Algorithm because the loss is convex, otherwise we can still sample according to the weights.

In contrast, vanilla weighted majority algorithm achieves $\tilde{O}(\sqrt{T})$ regret only over the whole interval $[1, T]$, and we improve upon the previous best result $\tilde{O}(\sqrt{t-s})$ Daniely et al. (2015) Jun et al. (2017). The proof of Theorem 2 can be found in the appendix.

### 3.1 OPTIMAL ADAPTIVE REGRET WITH ADAPTIVE GRADIENT METHODS

In this subsection, we show how to achieve full-matrix adaptive regret bounds by using Adagrad as experts as an application of Theorem 2, together with other extensions. We note that this reduction is general, and can be applied with any adaptive gradient method that has a regret guarantee, such as Adam or Adadelta.

Theorem 2 bounds the regret $R_1$ of the multiplicative weight part, while the total regret is $R_0 + R_1$. To get the optimal total regret bound, we only need to find an expert algorithm that also haves the optimal full-matrix regret bound matching that of $R_1$. As a result, we choose Adagrad as our expert algorithm $\boldsymbol{A}$, and prove regret bounds for both full-matrix and diagonal-matrix versions.

**Full-matrix adaptive regularization**

**Corollary 4** (Full-matrix Adaptive Regret Bound). *Under assumptions 1 and 2, when Adagrad is used as the blackbox $\mathcal{A}$, the total regret Regret$(I)$ of the multiplicative weight algorithm in Algorithm 1 satisfies that for any interval $I = [s, t]$,*

$$Regret(I) = O\left(D \log(T) \max\left\{G\sqrt{\log(T)}, d^{\frac{1}{2}}\sqrt{\min_{H \in \mathcal{H}} \sum_{\tau=s}^{t} \|\nabla_\tau\|_H^{*2}}\right\}\right)$$

**Remark 5.** We notice that the $\log(T)$ overhead is brought by the use of $S$ and Cauchy-Schwarz. We remark here that by replacing $S$ with the set of all sub-intervals, we can achieve an improved bound with only a $\sqrt{\log(T)}$ overhead using the same analysis. On the other hand, such improvement in regret bound is at the cost of efficiency, that each round we need to make $\Theta(T)$ computations.

**Diagonal-matrix adaptive regularization**   If we restrict our expert optimization algorithm to be diagonal Adagrad, we can derive a similar guarantee for the adaptive regret.

**Corollary 6.** *Under assumptions 1 and 2, when diagonal Adagrad is used as the blackbox $\mathcal{A}$, the total regret Regret$(I)$ of the multiplicative weight algorithm in Algorithm 1 satisfies that for any interval $I = [s, t]$,*

$$Regret(I) = \tilde{O}\left(D_\infty \sum_{i=1}^{d} \|\nabla_{s:t,i}\|_2\right)$$

Here $\nabla_{s:t,i}$ denotes the $ith$ coordinate of $\sum_{\tau=s}^{t} \nabla_\tau$.

## 4 EXPERIMENTS

In this section, we demonstrate empirical effectiveness of the proposed framework for online and offline learning scenarios. For online learning experiment, we consider a simulated data distribution shift setting using CIFAR-10. For offline supervised learning, we experimented on standard benchmarks in vision and natural language processing domains.

### 4.1 ONLINE EXPERIMENTS

**experiment setup**: Our simulated online experiment is designed to assess robustness to unforeseen data distribution changes during training. Algorithms do not know in advance whether or when the data shift will happen. We design this online data distribution shift with the CIFAR-10 dataset. We partition the CIFAR-10 dataset into two non-overlapping groups with five classes each. We denote $D_1$ as the distribution for the first subset of data $\{X_1, Y_1\}$ and $D_2$ for the other subset of data $\{X_2, Y_2\}$. Specifically, the two subsets of data we used in our implementation have categories {dog, frog, horse, ship, truck} and {airplane, automobile, bird, cat, deer}. We shift the data from $D_1$ to $D_2$ at iteration 17,000 out of a total of 25,600 training iterations. We choose this transition time point because empirically all baselines have stable performance at this point, which permits a fair comparison when the data shift occurs. We use the ResNet-18 model for all experiments under this online setup. Since each subset of data only contains 5 classes, we modified the model's last layer corresponding.

**baselines**: We compare our learning rate adaptation framework with different combinations of off-the-shelf learning rate schedulers and optimizers from the optax library. To ensure a fair comparison, we well-tuned the hyperparameters associated with each of the baseline learning rate schedule $\times$ optimizer combinations. Specifically, our baseline learning rate schedulers include constant learning rate, cosine annealing, exponential decay, and warmup with cosine annealing. Our baseline optimizers include SGD, AdaGrad, and Adam. In total, we have 12 learning rate scheduler $\times$ optimizer pairs for baseline experiments. We report detailed hyperparameter choices for each baseline in the appendix.

**evaluation metrics**: We evaluate our method and baselines using three performance metrics:

- *post-shift local accuracy*: the average evaluation accuracy during a specified window starting at the beginning of the data distribution shift. We consider three window sizes: 100, 500, and 1000 iterations. This metric is used to measure the robustness of algorithms immediately after the data distribution change.
- *pre-shift accuracy*: the maximum evaluation accuracy prior to the data distribution shift.
- *post-shift accuracy*: the maximum evaluation accuracy after the data distribution shift.

**implementation**: We follow Algorithm 1 for SAMUEL implementation under the online setup. Our SAMUEL framework admits any choice of black-box OCO algorithms; for our online experiment we use Adagrad. Each expert is an Adagrad optimizer with a specific external learning rate multiplier. The total number of training iterations is 25,600 and we specify the smallest geometric interval to

|  | constant lr | | | cosine annealing | | |
| --- | --- | --- | --- | --- | --- | --- |
|  | SGD | AdaGrad | Adam | SGD | AdaGrad | Adam |
| avg acc. (window100) | 62.44±0.93 | 63.02±1.84 | 69.39±0.41 | 71.51±1.77 | 76.71±0.24 | 72.35±1.54 |
| avg acc. (window500) | 73.57±0.47 | 77.02±0.98 | 84.41±0.19 | 82.14±0.45 | 84.13±0.41 | 85.87±0.32 |
| avg acc. (window1000) | 81.33±0.25 | 81.34±0.77 | 87.55±0.14 | 85.05±0.32 | 86.95±0.33 | 88.72±0.16 |
| pre-shift acc. | 96.29±0.04 | 96.26±0.12 | 96.87±0.05 | 97.06±0.05 | 97.41±0.00 | 97.35±0.12 |
| post-shift acc. | 93.87±0.23 | 93.49±0.17 | 94.27±0.02 | 92.80±0.45 | 94.02±0.15 | 94.32±0.16 |

| **SAMUEL (ours)** | warmup cosine annealing | | | exponential decay | | |
| --- | --- | --- | --- | --- | --- | --- |
|  | SGD | AdaGrad | Adam | SGD | AdaGrad | Adam |
| **79.73±0.98** | 71.48±0.64 | 74.17±1.87 | 67.13±1.48 | 69.64±0.77 | 74.68±0.57 | 69.71±0.83 |
| **87.31±0.16** | 83.27±0.40 | 84.23±0.36 | 83.00±0.43 | 78.83±0.58 | 82.42±0.16 | 82.14±0.36 |
| **89.21±0.05** | 86.12±0.21 | 86.81±0.15 | 86.49±0.22 | 81.96±0.44 | 85.06±0.12 | 85.66±0.27 |
| **97.47±0.13** | 97.26±0.10 | 97.06±0.14 | 96.88±0.09 | 96.88±0.03 | 97.22±0.14 | 97.27±0.02 |
| **94.79±0.23** | 93.27±0.07 | 93.25±0.12 | 93.13±0.43 | 90.52±0.22 | 91.44±0.27 | 92.77±0.32 |

Table 2: Five accuracy metrics (%) for SAMUEL and baseline methods under online data distribution shift setup. Standard deviation is computed using three runs with different random seeds.

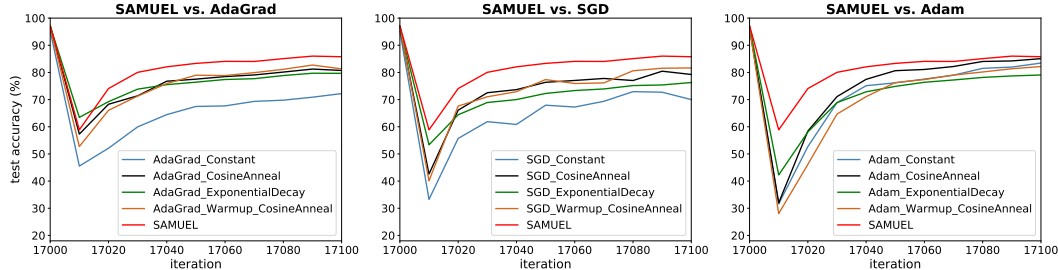

Figure 1: Behavior comparison following data distribution shift. Each subplot compares SAMUEL with an optimizer paired with different learning rate schedulers. We focus on a window of size 100 iterations post data distribution shift. SAMUEL systematically recovers fastest from data change and has a leading test accuracy throughout the window. The confidence band for each trace is the standard deviation computed across three different random seeds.

have length of 200 iterations. In total, the geometric intervals specified in Algorithm 1 have 8 different lengths, and therefore at each training iteration, experts are running on 8 different geometric intervals. Furthermore, we provide five learning rate candidates [0.05, 0.1, 0.25, 0.5, 1] to SAMUEL. In total 40 experts run at each training iteration. All experiments were carried out on TPU-V2 hardware with training batch size of 512.

**results**: We report the quantitative scores under five evaluation metrics of our algorithm and baselines in Table 2. We find that SAMUEL surpasses all baselines for every performance metric we considered. Although a number of baselines, such as Adagrad with cosine annealing, Adam with cosine annealing, and SGD with warmup cosine annealing, have comparable pre-shift test accuracy to SAMUEL, SAMUEL's ability to adaptively select the learning rate multiplier confers robustness to unforeseen changes in data distribution. This is unsurprising, given that typical off-the-shelf learning rate schedulers give a deterministic learning rate multiplier function across training and are therefore prone to suffering from data distribution changes. We also compare the qualitative behaviors of our algorithm and baselines within a 100-iteration window after the data distribution change in Figure 1. It is clear from the plots that SAMUEL recovers faster than baselines. Furthermore, SAMUEL consistently maintains a higher test accuracy throughout the window.

## 4.2 Offline Experiments

**experiment setup**: We experiment with popular vision and language tasks to demonstrate SAMUEL's ability in selecting optimal learning rates on-the-fly without hyperparameter tuning. The tasks conducted are image classification on CIFAR-10 and ImageNet, and sentiment classification on SST-2. We use ResNet-18 for CIFAR-10, ResNet-50 for ImageNet, and LSTM for SST-2.

**baseline**: We use the step learning rate scheduler as baseline, which is a commonly used off-the-shelf scheduler. We specifically use a three-phase schedule where we fix the two step transition points based on heuristics and provide five candidate learning rates to each phase. An exhaustive search thus yields a total of 125 different schedules.

**implementation**: We adjusted Algorithm 1 to be computationally efficient. Instead of running experts for each of the $\log T$ geometric intervals, we take a fixed number of experts (five total experts for these experiments, with one candidate learning rate per expert) with exponential decay factor on the history. Unlike Algorithm 1 where experts are initialized at the start of each geometric interval, we initialize experts at the step transition points. We introduce a parameter $\alpha$ that determines the effective memory length: $x_{t+1} = x_t - \frac{\eta}{\sqrt{\epsilon I + \sum_{\tau=1}^{t} \alpha^{t-\tau} \nabla_\tau \nabla_\tau^\top}} \nabla_t$. A fixed interval with different $\alpha$s can be seen as a "soft" version of the geometric intervals in Algorithm 1. All experiments were conducted on TPU-V2 hardware. We provide pseudo-code for the implementation in the appendix.

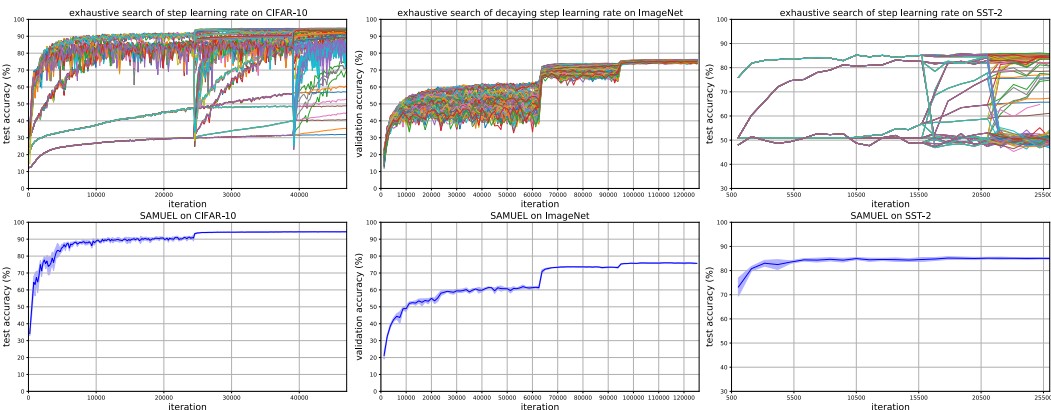

Figure 2: Comparison of exhaustive searched step learning rate schedule (top) and SAMUEL (bottom) on CIFAR-10, ImageNet and SST-2.

**CIFAR-10**: We compare a ResNet-18 model trained with SAMUEL to ResNet-18 trained with Adagrad using brute-force searched step learning rate schedules. We process and augment the data following He et al. (2016). For training, we use a batch size of 256 and 250 total epochs. We fix the learning rate transition point at epoch 125 and 200, and provide five candidate learning rates $\{0.0001, 0.001, 0.01, 0.1, 1\}$ for each region. Thus an exhaustive search yields 125 different schedules for the baseline. For a fair comparison, we adopt the same learning rate changing points for our method. We compare the test accuracy curves of the baselines and our methods in Fig.2. The left plot in Fig.2 displays 125 runs using Adagrad for each learning rate schedule, where the highest accuracy is 94.95%. A single run of SAMUEL achieves 94.76% with the same random seed (average 94.50% across 10 random seeds), which ranks in the top 3 of 125 exhaustively searched schedules.

**ImageNet**: We continue examining the performance of SAMUEL on the large-scale ImageNet dataset. We trained ResNet-50 with exhaustive search of learning rate schedules and compare with SAMUEL. We also consider a more practical step learning rate scheduling scheme where the learning rate decays after each stepping point. Specifically, the candidate learning rates are $\{0.2, 0.4, 0.6, 0.8, 1.0\}$ in the first phase, and decay by $10\times$ when stepping into the next phase. We set the stepping position at epoch 50 and 75 in a total of 100 training epochs. We adopted the training pipeline from Heek et al. (2020). For both baselines and SAMUEL, we used the SGD optimizer with nesterov momentum of 0.9 and training batch size of 1024. The second column of Fig.2 displays the comparison of the exhaustive search baseline (top) to SAMUEL (bottom). The best validation accuracy out of exhaustively searched learning rate schedules is 76.32%. SAMUEL achieves 76.22% in a single run (average 76.15% across 5 random seeds). Note that 76.22% is near-SOTA given the model architecture.

**SST-2**: We conduct experiments on the Stanford Sentiment Treebank (SST-2) dataset. We adopt the pipeline from (Heek et al., 2020) for pre-processing the SST-2 dataset and train a simple bi-directional

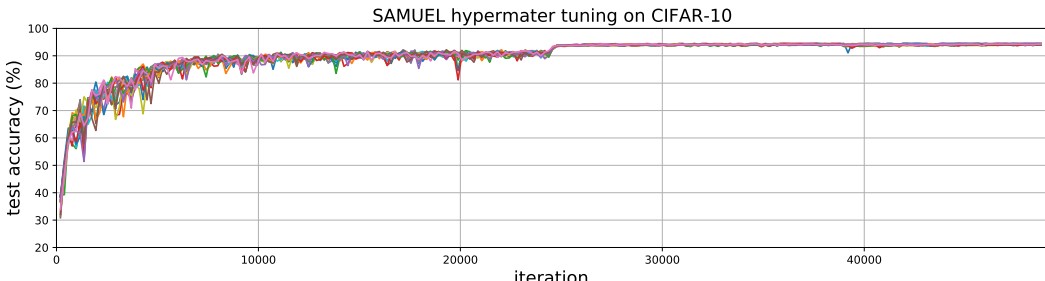

Figure 3: stability study of SAMUEL with different hyperparameters.

LSTM text classifier. We set the learning rate step transitions at epoch 15 and 20 in a total 25 training epochs. For both baseline and our algorithm, we use SGD with momentum of 0.9 and additive weight decay of 3e-6 with training batch size of 64. The learning rate schedule setting is the same as that of CIFAR-10. The right column of Fig. 2 shows that the best accuracy of exhaustive search is 86.12%, and the accuracy of SAMUEL using the same seed is 85.55% (average 85.58% among 10 different random seeds).

**stability of SAMUEL** : We demonstrate the stability of SAMUEL to hyperparameter tuning. Since our algorithm will automatically select the optimal learning rate, the only tunable hyperparameters are the number of multiplicative weight factor $\eta$ and the quantity of history decaying factors, $\alpha$. We conduct 18 trials with different hyperparameter combinations and display the test accuracy curves in Fig.3. Specifically, we consider the quantity of decaying factors $\alpha$ with values $\{2, 3, 6\}$ and $\{5, 10, 15, 20, 25, 30\}$ number of $\eta$ . As Fig.3 shows, all trials in SAMUEL converge to nearly the same final accuracy regardless of the exact hyperparameters.

**computation considerations**: A table of runtime comparison is provided in the appendix. As described in the implementation section, SAMUEL here has five experts in total, which incurs five times more compute than one single run of the baseline. Nevertheless, this is a dramatic improvement over brute-force hyperparameter sweeping of learning rate schedulers. For the step learning rate scheduler we experimented with, SAMUEL is 25 times more computationally efficient than tuning the scheduler with grid search. In addition, experts can be fully parallelized across different acceleration devices. It is expected that the run time of SAMUEL would approach that of a single run of the baseline with efficient implementation.

## 5 CONCLUSION

In this paper we study adaptive gradient methods with local guarantees. The methodology is based on adaptive online learning, in which we contribute a novel twist on the multiplicative weight method that we show has better adaptive regret guarantees than state of the art. This, combined with known results in adaptive gradient methods, gives an algorithm SAMUEL with optimal full-matrix local adaptive regret guarantees. We demonstrate the effectiveness and robustness of SAMUEL in experiments, where we show that SAMUEL can automatically adapt to the optimal learning rate and achieve better task accuracy in online tasks with distribution shifts. For offline tasks, SAMUEL consistently achieves comparable accuracy to an optimizer with fine-tuned learning rate schedule, using fewer overall computational resources in hyperparameter tuning.

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

# A APPENDIX

## A.1 PROOF OF THEOREM 2

*Proof.* We define the pseudo weight $\tilde{w}_\tau(I, q) = w_\tau(I, q)/\eta_{I,q}$ for $\tau \leq t$, and for $\tau > t$ we just set $\tilde{w}_\tau(I, q) = \tilde{w}_t(I, q)$. Let $\tilde{W}_\tau = \sum_{I \in S(\tau), q} \tilde{w}_\tau(I, q)$, we are going to show the following inequality

$$\tilde{W}_\tau \leq \tau(\log(\tau) + 1)\log(dTD^2G^2)\log(T) \tag{1}$$

We prove this by induction. For $\tau = 1$ it follows since on any interval $[1, t]$ the number of experts is exactly the number of possible $q$s, and the number of intervals $[1, t] \subset S$ is $O(\log(T))$. Now we assume it holds for all $\tau' \leq \tau$. We have

$$\tilde{W}_{\tau+1} = \sum_{I \in S(\tau+1), q} \tilde{w}_{\tau+1}(I, q)$$

$$= \sum_{I=[\tau+1, t] \in S(\tau+1), q} \tilde{w}_{\tau+1}(I, q) + \sum_{I=[s, t], s \leq \tau \in S(\tau+1), q} \tilde{w}_{\tau+1}(I, q)$$

$$\leq \log(\tau+1)\log(dTD^2G^2)\log(T) + 1 + \sum_{I=[s, t], s \leq \tau \in S(\tau+1), q} \tilde{w}_{\tau+1}(I, q)$$

$$= \log(\tau+1)\log(dTD^2G^2)\log(T) + 1 + \sum_{I=[s, t], s \leq \tau \in S(\tau+1), q} \tilde{w}_\tau(I, q)(1 + \eta_{I,q}r_\tau(I))$$

$$\leq \log(\tau+1)\log(dTD^2G^2)\log(T) + 1 + \tilde{W}_\tau + \sum_{I \in S(\tau), q} w_\tau(I, q)r_\tau(I)$$

$$\leq (\tau+1)(\log(\tau+1) + 1)\log(dTD^2G^2)\log(T) + \sum_{I \in S_\tau, q} w_\tau(I, q)r_\tau(I)$$

We further show that $\sum_{I \in S(\tau), q} w_\tau(I, q)r_\tau(I) \leq 0$:

$$\sum_{I \in S(\tau), q} w_\tau(I, q)r_\tau(I) = W_\tau \sum_{I \in S(\tau), q} p_\tau(I, q)(\ell_\tau(x_\tau) - \ell_\tau(x_\tau(I, q)))$$

$$\leq W_\tau \sum_{I \in S(\tau), q} p_\tau(I, q)\left(\sum_{J \in S(\tau), q} w_\tau(J, q)\ell_\tau(x_\tau(J, q))/W_\tau - \ell_\tau(x_\tau(I, q))\right)$$

$$= 0$$

which finishes the proof of induction.

Based on this, we proceed to prove that for any $I = [s, t] \in S$,

$$\sum_{\tau=s}^{t} r_\tau(I) = O\left(\sqrt{\log(T)} \max\left\{DG\sqrt{\log(T)}, \sqrt{\sum_{\tau=s}^{t}(\nabla_\tau^\top(x_\tau - x_\tau(I)))^2}\right\}\right)$$

By inequality 1, we have that

$$\tilde{w}_{t+1}(I, q) \leq \tilde{W}_{t+1} \leq (t+1)(\log(t+1) + 1)\log(dTD^2G^2)\log(T)$$

Taking the logarithm of both sides, we have

$$\log(\tilde{w}_{t+1}(I, q)) \leq \log(t+1) + \log(\log(t+1) + 1) + \log(\log(dTD^2G^2)) + \log(\log(T))$$

Recall the expression

$$\tilde{w}_{t+1}(I, q) = \prod_{\tau=s}^{t}(1 + \eta_{I,q}r_\tau(I))$$

By using the fact that $\log(1 + x) \geq x - x^2, \forall x \geq -1/2$ and

$$|\eta_{I,q}r_\tau(I)| \leq \frac{1}{4GD}\|x_\tau - x_\tau(I, q)\|_2 G \leq 1/2$$

we obtain for any $q$

$$\log(\tilde{w}_{t+1}(I,q)) \geq \sum_{\tau=s}^{t} \eta_{I,q} r_\tau(I) - \sum_{\tau=s}^{t} \eta_{I,q}^2 r_\tau(I)^2$$

Now we upper bound the term $\sum_{\tau=s}^{t} r_\tau(I)^2$. By convexity we have that $r_\tau(I) = \ell_\tau(x_\tau) - \ell_\tau(x_\tau(I)) \leq \nabla_\tau^\top (x_\tau - x_\tau(I))$, hence

$$\sum_{\tau=s}^{t} r_\tau(I) \leq \frac{4\log(T)}{\eta_{I,q}} + 4\eta_{I,q} \sum_{\tau=s}^{t} (\nabla_\tau^\top (x_\tau - x_\tau(I)))^2$$

The next step is to upper bound the term $\nabla_\tau^\top (x_\tau - x_\tau(I))$. By Hölder's inequality we have that $\nabla_\tau^\top (x_\tau - x_\tau(I)) \leq \|\nabla_\tau\|_{H^{-1}} \|x_\tau - x_\tau(I)\|_H$ for any $H$. As a result, we have that for any $H$ which is PSD and $tr(H) \leq d$,

$$(\nabla_\tau^\top (x_\tau - x_\tau(I)))^2 \leq \nabla_\tau^\top H^{-1} \nabla_\tau \|x_\tau - x_\tau(I)\|_H^2 \leq \nabla_\tau^\top H^{-1} \nabla_\tau 4D^2 d$$

where $\|x_\tau - x_\tau(I)\|_H^2 \leq 4D^2 d$ is by elementary algebra: let $H = V^{-1} M V$ be its diagonal decomposition where $B$ is a standard orthogonal matrix and $M$ is diagonal. Then

$$\begin{aligned}
\|x_\tau - x_\tau(I)\|_H^2 &= (x_\tau - x_\tau(I))^\top H (x_\tau - x_\tau(I)) \\
&= (V(x_\tau - x_\tau(I)))^\top MV(x_\tau - x_\tau(I)) \\
&\leq (V(x_\tau - x_\tau(I)))^\top dIV(x_\tau - x_\tau(I)) \\
&\leq 4D^2 d
\end{aligned}$$

Hence

$$\sum_{\tau=s}^{t} r_\tau(I) \leq \frac{4\log(T)}{\eta_{I,q}} + 4\eta_{I,q} D^2 d \min_H \sum_{\tau=s}^{t} \nabla_\tau^\top H^{-1} \nabla_\tau$$

The optimal choice of $\eta$ is of course

$$4\sqrt{\frac{\log(T)}{D^2 d \min_H \sum_{\tau=s}^{t} \nabla_\tau^\top H^{-1} \nabla_\tau}}$$

When $D^2 d \min_H \sum_{\tau=s}^{t} \nabla_\tau^\top H^{-1} \nabla_\tau \leq 64G^2 D^2 \log(T)$, $\eta_{I,1}$ gives the bound $O(GD\log(T))$. When $D^2 d \min_H \sum_{\tau=s}^{t} \nabla_\tau^\top H^{-1} \nabla_\tau > 64G^2 D^2 \log(T)$, there always exists $q$ such that $0.5\eta_{I,q} \leq \eta \leq 2\eta_{I,q}$ by the construction of $q$ so that the regret $R_1(I)$ is upper bounded by

$$O\left(D\sqrt{\log(T)} \max\left\{G\sqrt{\log(T)}, d^{\frac{1}{2}} \sqrt{\min_{H\in\mathcal{H}} \sum_{\tau=s}^{t} \nabla_\tau^\top H^{-1} \nabla_\tau}\right\}\right) \tag{2}$$

Now we have proven an optimal regret for any interval $I \in S$, it's left to extend the regret bound to any interval $J$. We show that by using Cauchy-Schwarz, we can achieve the goal at the cost of an additional $\sqrt{\log(T)}$ term. We need the following lemma from Daniely et al. (2015):

**Lemma 7** (Lemma 5 in Daniely et al. (2015)). *For any interval $J$, there exists a set of intervals $S^J$ such that $S^J$ contains only disjoint intervals in $S$ whose union is exactly $J$, and $|S_J| = O(\log(T))$*

We now use Cauchy-Schwarz to bound the regret:

**Lemma 8.** *For any interval $J$ which can be written as the union of $n$ disjoint intervals $\cup_i I_i$, its regret $Regret(J)$ can be upper bounded by:*

$$Regret(J) \leq \sqrt{n \sum_{i=1}^{n} Regret(I_i)^2}$$

*Proof.* The regret over $J$ can be controlled by $Regret(J) \leq \sum_{i=1}^{n} Regret(I_i)$. By Cauchy-Schwarz we have that

$$(\sum_{i=1}^{n} Regret(I_i))^2 \leq n \sum_{i=1}^{n} Regret^2(I_i)$$

which concludes our proof. $\qquad \square$

We can now upper bound the regret $R_1(J)$ using Lemma 8, replacing $Regret$ by $R_1$ and $n$ by $|S_J| = O(\log(T))$. For any interval $J$, its regret $R_1(J)$ can be upper bounded by:

$$R_1(J) \leq \sqrt{|S_J| \sum_{I \in S_J} R_1(I)^2}$$

Combining the above inequality with the upper bound on $R_1(I)$ 2, we reach the desired conclusion.
$\qquad \square$

## A.2 PROOF OF COROLLARY 4

*Proof.* Using Theorem 2 we have that $R_1(I)$ is upper bounded by

$$R_1(I) = O\left(D \log(T) \max\left\{ G\sqrt{\log(T)}, d^{\frac{1}{2}} \sqrt{\min_{H \in \mathcal{H}} \sum_{\tau=s}^{t} \|\nabla_\tau\|_H^{*2}} \right\}\right)$$

Because on each interval $J \in S$, one of the Adagrad experts achieve the bound

$$R_0(J) = O\left(D d^{\frac{1}{2}} \sqrt{\min_{H \in \mathcal{H}} \sum_{\tau=s}^{t} \|\nabla_\tau\|_H^{*2}}\right)$$

For any interval $I$, using the result from Daniely et al. (2015) (Lemma 7) and Lemma 8 by replacing $Regret$ by $R_0$, it follows

$$R_0(I) = O\left(D \sqrt{\log(T)} d^{\frac{1}{2}} \sqrt{\min_{H \in \mathcal{H}} \sum_{\tau=s}^{t} \|\nabla_\tau\|_H^{*2}}\right)$$

Combining both bounds give the desired bound on $Regret(I)$. $\qquad \square$

## A.3 PROOF OF COROLLARY 6

*Proof.* The proof is almost identical to that of the previous corollary, observing that the regret $R_0(I)$ is $\tilde{O}(D_\infty \sum_{i=1}^{d} \|\nabla_{s:t,i}\|_2)$ due to Duchi et al. (2011), and the regret $R_1(I)$ remains $\tilde{O}(D\sqrt{\min_{H \in \mathcal{H}} \sum_{\tau=s}^{t} \nabla_\tau^\top H^{-1} \nabla_\tau})$, which is upper bounded by $\tilde{O}(D_\infty \sum_{i=1}^{d} \|\nabla_{s:t,i}\|_2)$. $\qquad \square$

## A.4 BASELINE HYPERPARAMETERS FOR ONLINE EXPERIMENTS

Here we report the hyperparmeters used in the baseline learning rate schedulers in the online experiments. We use the off-the-shelf learning rate schedulers from the optax library. Please refer to the optax documentation for the specific meaning of the parameters.

AdaGrad

- constant learning rate: learning rate 0.2.
- cosine annealing: init value = 0.2, decay steps = 25600, alpha = 0.
- warmup with cosine annealing: init value = 1e-5, peak value = 0.15, warmup steps = 1000, end value = 0.
- exponential decay: init value = 0.35, transition steps= 3000, decay rate = 0.5.

SGD

- constant learning rate: learning rate 0.15.
- cosine annealing: init value = 0.3, decay steps = 25600, alpha = 0.
- warmup with cosine annealing: init value = 1e-5, peak value = 0.5, warmup steps = 1000, end value = 0.
- exponential decay: init value = 0.6, transition steps= 3000, decay rate = 0.5.

Adam

- constant learning rate: learning rate 0.001.
- cosine annealing: init value = 0.001, decay steps = 25600, alpha = 0.
- warmup with cosine annealing: init value = 1e-5, peak value = 0.005, warmup steps = 1000, end value = 0.
- exponential decay: init value = 0.005, transition steps= 3000, decay rate = 0.5.

## A.5 COMPUTE COMPARISON FOR OFFLINE EXPERIMENTS

We report the compute resource consumption of both baselines and SAMUEL from the offline experiments. We run experts sequentially and the running time of our algorithm is longer than the baselines. With more efficient implementation and parallelizing each expert across TPU devices, it is expected the running time of SAMUEL would approach the running time of the baseline algorithm.

| CIFAR-10 | device config | runtime (m) | grid-search cost (trials) | runtime per expert (m) | total TPU hours |
|---|---|---|---|---|---|
| **baseline** | 4TPU | 11 | 125 | 11 | 91.6 |
| **SAMUEL** | 4TPU | 66 | 1 | 13.2 | 4.4 |
| ImageNet | | | | | |
| **baseline** | 4TPU | 254 | 125 | 254 | 2116.6 |
| **SAMUEL** | 16TPU | 794 | 1 | 158.8 | 211.7 |
| SST-2 | | | | | |
| **baseline** | 1TPU | 12 | 125 | 12 | 25 |
| **SAMUEL** | 4TPU | 25 | 1 | 5 | 1.6 |

Table 3: compute comparison

## A.6 PSEUDOCODE FOR OFFLINE EXPERIMENTS

---

**Algorithm 2** SAMUEL experiment pseudocode

---

1: Input: AdaGrad optimizer $\boldsymbol{A}$, constant Q, a set of learning rates $\{1, 0.1, 0.001, 0.0001, 0.00001\}$, reinitialize frequency K.
2: Initialize: for each learning rate $i \in S$, a copy of $\boldsymbol{A}_i$.
3: Set $\eta_{i,q} = \frac{1}{2^q}$ for $q \in [1, Q]$.
4: Initialize $w_1(i, q) = \min\{1/2, \eta_{I,q}\}$. Initialize NN params $x_0$
5: **for** $\tau = 1, \ldots, T$ **do**
6:      Let updated NN params $x_\tau(i, q) = \boldsymbol{A}_i(\tau)$
7:      Let $W_\tau = \sum_{i,q} w_\tau(i, q)$.
8:      sample $x_\tau$ according to $w_\tau(i, q)/W_\tau$.
9:      Receive batch loss $\ell_\tau(x_\tau)$, define $r_\tau(i) = \ell_\tau(x_\tau) - \ell_\tau(x_\tau(i, q))$.
10:      For each $i$, update $w_{\tau+1}(i, q)$ as follows.

$$w_{\tau+1}(i, q) = w_\tau(i, q)(1 + \eta_{i,q} r_\tau(i))$$

11:      **if** $\tau \% K = 0$ **then**
12:          Re-initialize $w_\tau(i, q) = \min\{1/2, \eta_{I,q}\}$
13:          All copies $\boldsymbol{A}_i$ start from NN params $x_\tau$
14:      **end if**
15: **end for**

---

