# OpenReview forum: "Adaptive Gradient Methods with Local Guarantees"
_ICLR.cc/2023/Conference — Submitted to ICLR 2023_

### Official Review · Reviewer_vT2z · 2022-10-20

**Confidence:** 4
**Correctness:** 1
**Technical Novelty And Significance:** 3
**Empirical Novelty And Significance:** 3
**Recommendation:** 5

**Clarity, Quality, Novelty And Reproducibility:**

***Clarity***
The paper is clear. Nevertheless, for the reader to correctly understand the connection of this paper with existing work, it should be explicitly and clearly stated that the first part of the proof of Theorem 2, which occupies more than one page, simply follows the proof of Daniely et al.

***Quality***
The presentation is fine. Below are some minor flaws.
- Algorithm 1: Should $A_I (\tau)$ be $A_{I, q} (\tau)$?
- p. 5: "Weighted majority" should be "multiplicative weight," too keep consistency of the terms.
- Mathematical operators like "tr" should be in roman.
- Please adjust the sizes of the brackets in the first two equations on p. 2.

***Novelty***
The idea of developing an adaptive version of AdaGrad is novel. The regret bound of SAOL in Theorem 2 is the main technical breakthrough, but its correctness is in question; see my explanation above.

This is minor: The second last paragraph of Section 1.1 and the second paragraph on p. 5 seem to be a novelty statement for SAOL of Daiely et al. and not for this paper.

***Reproducibility***
The pseudo code for the numerical experiments is provided in Algorithm 2 in the appendix.

**Strength And Weaknesses:**

***Strength***
The idea is novel, simple, and seems to be effective in practice.

***Weaknesses***
1. I do not understand the proof of Theorem 2, the main result, in Appendix A. Perhaps I missed something. The proof consists of three parts, among which the second part is new and confusing.
    - The first part, until the first inequality on p. 13, just follows the proofs of Lemma 1 and Lemma 2 of Daniely et al.
    - The second part is new and confusing to me. It directly analyzes the regret, without referring to any intermediate result obtained in the first part. The meaning of the first part is then unclear.
    - Moreover, the second part starts with a regret bound of SAOL, which is the second inequality on p. 13. The regret bound is given without any proof and citation, but it is not obviously true to me.
    - The third part, starting from Lemma 7, is a directly application of the results by Daniely et al.
2. If I understand correctly, the result in this paper does not actually inherit the regret bound of AdaGrad. The regret bound in this paper has a $\sqrt{d}$ factor which is not present in the regret bound of AdaGrad (see, e.g., Theorem 5.6.1 of the second version of *Introduction to Online Convex Optimization* by Hazan).
3. This is minor: Why not define $D$ as the radius of the constraint set?



**Summary Of The Paper:**

Based on the strongly adaptive regret framework of Daniely et al., this paper proposes a strongly adaptive version of AdaGrad. The novelty seems to lie in an improved regret bound of SAOL, the meta algorithm of Daniely et al., which is similar to the regret bound of AdaGrad.

**Summary Of The Review:**

The idea is novel, simple, and seems to be empirically effective. However, the correctness of the theory is unclear to me, and the regret bound is indeed worse than that of AdaGrad by a $\sqrt{d}$ factor, which seems to be dismissed in the paper. Also, the connection between the proof of this paper and that by Daiely et al. is not addressed; this can mislead the reader about the paper's novelty. Therefore, I cannot suggest acceptance of this paper currently.

====

***After reading the author response***

The first two weaknesses were actually due to my carelessness. I am sorry.

However, the technical novelty seems to be limited and there is a novelty issue regarding the paper by Zhang et al. (NeurIPS 2021). So I cannot recommend acceptance of this paper.

Two suggestions to the authors:
1. Please highlight the novelty and address explicitly address what parts essentially follow existing proofs, to help the reader evaluate the novelty of the paper and its connection to existing literature.
2. Please make the dependence on dimension and possibly other problem parameters explicit in the big-O notations (e.g., in Table 1).

---

> ### Author Response · Authors · 2022-11-14
> **Response**
>
> We thank you for your valuable feedback! Regarding proof of Theorem 2, the second inequality on page 13 is a consequence of the first inequality on page 13 and convexity: moving the negative term from RHS to LHS in the first inequality and divide both sides by $\eta_{I,q}$, then using the convexity $r_{\tau}(I)\le \nabla_{\tau}^{\top}(x_{\tau}-x_{\tau}(I))$ gives the second inequality. We will add these omitted intermediate steps to make it more clear.
>
> Regarding the bound of Adagrad: there is also a $\sqrt{d}$ term in the bound of Adagrad, see section 1.2 in the original Adagrad paper. However, another potential issue is that due to the $\sqrt{d}$ term, the full-matrix Adagrad bound does not improve upon the second-order SGD bound, and the improved bounds are obtained only for the diagonal version, just as in the original Adagrad paper, please see response to reviewer 1.

---

> > ### Comment · Reviewer_vT2z · 2022-12-08
> > **Re: Response**
> >
> > Sorry for the late response.
> >
> > 1. Sorry for my careless review. The inequality is indeed simple and does not need further explanation. ***Please highlight the novelty and address explicitly address what parts essentially follow existing proofs, to help the reader evaluate the novelty of the paper and its connection to existing literature.***
> >
> > 2. Your explanation on the $\sqrt{d}$ term is correct. ***Please make the dependence on dimension and possibly other problem parameters explicit in the big-O notations (e.g., in Table 1).***
> >
> > 3. Now the results appear to be correct and reasonable. I like the simple and effective idea of making AdaGrad adaptive. However, the technical novelty seems to be limited and there is a novelty issue regarding the paper by Zhang et al. (NeurIPS 2021). So I cannot recommend acceptance of this paper.

---

### Official Review · Reviewer_fGkK · 2022-10-22

**Confidence:** 4
**Correctness:** 3
**Technical Novelty And Significance:** 2
**Empirical Novelty And Significance:** 1
**Recommendation:** 5

**Clarity, Quality, Novelty And Reproducibility:**

The writing was generally well-written, clear, and easy to follow throughout, and seemed somewhat reproducible since they provide pseudocode for their practical implementation and test on standard benchmark problems.

Novelty
---
I think the main contribution of this paper may have already been achieved. In particular, Theorem 1 of Zhang et. al. (2021) provides an algorithm which achieves
$$
R_I(u)\le \tilde O\left(\sqrt{\sum_{t\in I}\langle \nabla \ell_t(w_t), w_t-u\rangle^2}\right)
$$
for any $I\subseteq[1,T]$ and comparator $u$ in the (bounded) domain. This implies the desired $\tilde O\left(\min_{H\succ 0}\sqrt{\sum_{t\in I}\|\|\nabla \ell_t(w_t)\|\|^2_{\star,H}}\right)$ bound using the same arguments as found on page 13 of the submitted paper. Please correct me if I am mistaken.



References
---
[3] Zhang, Lijun, Guanghui Wang, Wei-Wei Tu, Wei Jiang, and Zhi-Hua Zhou. "Dual adaptivity: A universal algorithm for minimizing the adaptive regret of convex functions." Advances in Neural Information Processing Systems 34 (2021): 24968-24980.

**Strength And Weaknesses:**

Correctness
---
I have some concerns with the correctness of the analysis.

Following standard experts analysis, the regret over an interval $I$ is decomposed as the regret of an expert $A_k$ on the interval, $R_{A_k}(I)$, and the regret of the meta-algorithm on the interval $R_\mathit{meta}(I, A_k)$ for not choosing this expert. To get the right rate in the meta-algorithm, it's necessary to balance a $\frac{\log(T)}{\eta}$ and a $\eta\sum\langle\nabla\ell_t(w_t),w_t-u\rangle^2$ term, where $\eta$ is the scaling factor of AdaGrad's step-size (see [Duchi 2011, Algorithm 2](https://www.jmlr.org/papers/volume12/duchi11a/duchi11a.pdf#page=14)). Since the optimal $\eta^\star$ is unknown a priori, the proposed algorithm runs several instances of AdaGrad over a grid of values, and argue that some $\eta_k\approx\eta^\star$, so that the meta-algorithm has the correct bound.

The flaw then is that it is stated that $R_{A_k}(I)\le O(D\min_{H\succ 0}\sqrt{\sum \|\| \nabla\ell_t(w_t)\|\|^2_{\star,H}})$ since it is an instance of AdaGrad (i.e. [Corollary 11](https://www.jmlr.org/papers/volume12/duchi11a/duchi11a.pdf#page=16)). But this is not true for arbitrary $\eta$. Instead, one should be invoking [Theorem 7](https://www.jmlr.org/papers/volume12/duchi11a/duchi11a.pdf#page=13) to get
\begin{align*}
R_{A_k}(I)&\le O\left( \frac{\|\|x^{\star}\|\|^2\text{Trace}(G_t^{\frac{1}{2}})}{\eta_k}+\eta_k \text{Trace}(G_I^{\frac{1}{2}})\right)\\\\
&\le O\left( \frac{\|\|x^{\star}\|\|^2\sqrt{\min_H\sum_t\|\|\nabla\ell_t(w_t)\|\|^2_{\star,H}}}{\eta_k}+\eta_k\sqrt{\min_H\sum_t\|\|\nabla\ell_t(w_t)\|\|^2_{\star,H}}\right)
\end{align*}
where $G_I=\sum_{t\in I}\nabla \ell_t(w_t)\nabla\ell_t(w_t)^\top$, and the inequality applies [Lemma 15](https://www.jmlr.org/papers/volume12/duchi11a/duchi11a.pdf#page=31).
Now if we assume that $\eta_k\approx \eta^\star=\sqrt{\frac{\log(T)}{D^2d\min_H\sum_t\|\|\nabla\ell_t(w_t)\|\|^2_{\star,H}}}$ (from page 13), we seem to get something strange like
\begin{align*}
R_{A_k}(I)\le \tilde O\left(D^3\sqrt{d}\min_H\sum_t\|\|\nabla\ell_t(w_t)\|\|^2_{\star,H}\right).
\end{align*}


Aside from this, there were a few minor oddities:
- Why are we assuming $G\ge 1$ and $D\ge 1$? This is not standard shouldn't really be necessary either
- It seems strange that the number of experts $Q=4\log(dTG^2D^2)$ in each interval is tied to the Lipschitz constant and the size of the domain, I don't think i've seen this happen in any prior works and it seems like it could lead to infeasible computation even at short horizons. In general, It's sort of rare to see factors of $G$ and $D$ inside of a logarithm without some fudge factor $\epsilon$ to correct the "units". I suspect that this issue is actually related to the above assumption that $G\ge1$ and $D\ge 1$ and the number of experts in an interval should actually be more like $O(\log(T))$

The Algorithm
---
- To get this sort of bound, is there any reason you couldn't just apply the residual learning algorithm of Cutkosky 2020? Their approach needs the base algorithm to have the parameter-free property $R_I(0)=\sum_{t\in I}\langle\nabla\ell_t(w_t),w_t\rangle\le \epsilon$ for some $\epsilon$, but there are algorithms with full-matrix regret bounds that do this, such as the Matrix FreeGrad algorithm of Mhammedi 2020.

Experiments
---
- "As described in the implementation section, SAMUEL here has five experts in total, which incurs five times more compute than one single run of the baseline. Nevertheless, this is a dramatic improvement over brute-force hyperparameter sweeping of learning rate schedulers"
  - I'm confused by this.  The preceeding paragraph states "specifically, we consider the quantity of decaying factors α with values {2, 3, 6} and {5, 10, 15, 20, 25, 30} number of $\eta$"; does this not imply that SAMUEL takes either 5x, 10x, 15x, 20x, 25x, or 30x computation compared to the baseline? Moreover, since these hyperparameters were tuned, overall didn't SAMUEL use the equivalent computation of a baseline which had its hyperparameters tuned over $3*(5+10+15+20+25+30)=315$ values?
  - Similarly, at the start of the section it's stated that "We experiment with popular vision and language tasks to demonstrate SAMUEL’s ability in selecting optimal learning rates on-the-fly without hyperparameter tuning"; but it seems that $\eta$ and $\alpha$ were actually  tuned hyperparameters?
- Specific hyperparameter settings are given in the appendix, but as far as I can tell there's no mention of how the baselines' hyperparameters were actually tuned. Without this information I can't really tell if these are actually reasonable baselines to compare against.
- "We demonstrate the effectiveness and robustness of SAMUEL in experiments, where we show that SAMUEL can automatically adapt to the optimal learning rate"
  - This was never actually demonstrated, just that SAMUEL empirically outperformed the baselines, none of which necessarily are equivalent to using the optimal learning rate


References
---
[1] Mhammedi, Z., & Koolen, W. M. (2020). Lipschitz and comparator-norm adaptivity in online learning. In J. Abernethy, & S. Agarwal, Proceedings of Thirty Third Conference on Learning Theory (pp. 2858–2887). PMLR.

[2] Cutkosky, A. (2020). Parameter-free, dynamic, and strongly-adaptive online learning. In H. D. III, & A. Singh, Proceedings of the 37th International Conference on Machine Learning (pp. 2250–2259). PMLR.



**Summary Of The Paper:**

This paper proposes an algorithm for strongly-adaptive full-matrix regret of the form $R_I\le \tilde O(\min_{H\succ 0}\sqrt{\sum_{t\in I} \|\|\nabla\ell_t(w_t)\|\|^2_{\star,H}})$ for any interval $I\subseteq[1,T]$. The core idea is to apply the geometric covering intervals of Daniely 2015 with instances of full-matrix AdaGrad as the base learners. A simplified version of the algorithm is tested empirically in simple online and offline experiments.

**Summary Of The Review:**

Overall, I do not recommend accepting this paper in its current form. I am not confident in the correctness of the algorithm and analysis, which constitutes the main contributions of the paper, and a prior work (Zhang 2021) seems to achieve a strictly stronger result. The experimental results serve mostly as a sanity check, showing that a simplified version of their algorithm does something reasonable, but do not provide any novel insights in their own right.

---

> ### Author Response · Authors · 2022-11-14
> **Response**
>
> We thank you for your valuable feedback! Regarding your main concern, we believe it is a misunderstanding of the algorithm, and the proof is correct. The key point is that the meta MW algorithm and the adagrad experts don't share learning rates. The $\eta^*$ is only used for tuning the meta MW algorithm, and each expert gets the adagrad bound by itself independently.
>
> Regarding minor issues: 1, we can remove the assumptions on $G,D$ and instead replace them by $G+1,D+1$ in the bounds. 2, this is because the optimal $\eta^*$ depends on $G,D$, and to ensure we can find it using a binary search, the initial range should be set large enough which leads to $G,D$ within the logarithm.
>
> Regarding Zhang et al 18: unfortunately we were not aware of this paper, and it indeed implies our main theoretical result. Please see response to reviewer 1.
>
> We address the experiment concerns as follows:
>
> 1. Computation cost introduced by multiple $\eta$s in multiplicative weight update is negligible compared to the cost of training a deep neural network. Regarding the reviewer's confusion with ``specifically, we consider the quantity of decaying factors $\alpha$ with values {2, 3, 6} and {5, 10, 15, 20, 25, 30} number of ", here the sets of values are used in hyperparameter sweeping to test SAMUEL's robustness as shown in Figure 3. For a single trial, for example, with hyperparameter configuration of $\alpha =2$ and $\eta = 20$, SAMUEL's computation cost is about five times more than one trial of the baseline. This is because the major computation cost of SAMUEL comes from the five experts not $\eta$s. $\eta$s are only used in the multiplicative weight update, which is negligible compared to the cost of forward and backward propagation of a deep neural network.
>
> 2. Regarding your comments on "it seems that  and  were actually tuned hyperparameters", the hypereparameters sweeping section (see Figure 3) serves to demonstrate SAMUEL's robustness to hyperparameter choices. In other word, we sweep hyperparamters just to demonstrate that SAMUEL has very stable high performance for every combination of hyperparmeters tried.

---

> > ### Comment · Reviewer_fGkK · 2022-11-15
> > **Response to the Response**
> >
> > >  we believe it is a misunderstanding of the algorithm, and the proof is correct. The key point is that the meta MW algorithm and the adagrad experts don't share learning rates. The is only used for tuning the meta MW algorithm, and each expert gets the adagrad bound by itself independently
> >
> > Let me see if I understand correctly:
> > For any arbitrary expert $\mathcal{A}\_{I,i}$ in interval $I$ and any comparator $u$ we can write
> > \begin{align*}
> > R_I(u)
> > &=
> >     \sum_{t\in I}\ell_t(x_t)-\ell_t(u)\\\\
> > &=
> >     \sum_{t\in I}\ell_t(x_{t}(I,i))-\ell_t(u)+\sum_{t\in I}\ell_t(x_{t})-\ell_t(x_{t}(I,i))
> > \end{align*}
> > The first sum is the regret of the algorithm $\mathcal{A}\_{I,i}$ against
> > comparator $u$, where $\mathcal{A}\_{I,i}$ is an instance of AdaGrad, and the second term is essentially the regret of the MW
> > algorithm against a comparator that places all the probability on expert $\mathcal{A}\_{I,i}$.
> >
> > You're saying that the expert $\mathcal{A}\_{I,i}$ doesn't actually scale its step-sizes by $\eta_i$, so that in fact for all $i\in Q$, $\mathcal{A}\_{I,i}$ are exactly the same algorithm?
> > If this is the case, then yes I agree that all the experts will have the desired regret, since now you can argue that they're all
> > running AdaGrad with $\eta=\frac{D}{\sqrt{2}}$ and apply Duchi's corollary 11.
> > I think the presentation here is a bit confusing: why should we need copies of
> > the algorithm at all if they're all precisely the same algorithm and output an identical point? Typically in these tuning-via-experts arguments
> > the copy $\mathcal{A}\_{i}$ denotes the algorithm using step-size $\eta_i$ and outputs a distinct point $x_{t,i}$, but here $\mathcal{A}\_{I,i}=\mathcal{A}\_I$ and $x\_{t}(I,i)=x_t(I)$ for any $i\in Q$ which is a bit confusing because of the redundance of it
> >
> > On the other hand, if this is the case and all experts are just a copy of AdaGrad
> > then $x_{t}(I,i)=x_{t}(I,j)$ for any $i,j\in Q$. If we additionally have $x_t(I,i)=x_t(J,q)$
> > for $I,J\in S(t)$, then there would be no reason to add the copies of $\mathcal{A}$, since we would have
> > $x_t=\sum_{I\in S(t),q} p_t(I,q)x_t(I,q)=x_t(I,i)\sum_{I,q}p_{t}(I,q)=x_{t}(I,i)$ for arbitrary $i\in Q$
> > regardless. Otherwise, if $x_t(J,i)\ne x_t(I,i)$ for $I\ne J$, then
> > on page 12 we have
> > \begin{align*}
> > \sum_{I\in S(t),q}w_t(I,q)r_t(I)
> > &=
> > W_t\sum_{I\in S(t),q}p_t(I,q)(\ell_t(x_t)-\ell_t(x_{t}(I,q)))\\\\
> > &\le
> > W_t\sum_{I\in S(t),q}p_t(I,q)\left[\sum_{J\in S(t), i}\left(\frac{w_t(J, i)}{W_t}\ell_t(x_{t}(J,i))\right)-\ell_t(x_{t}(I,q))\right]\\\\
> > &=
> > W_t\sum_{I\in S(t),q}p_t(I,q)\left[\sum_{J\in S(t)}\left(\frac{\sum_{i\in Q}w_t(J, i)}{W_t}\right)\ell_t(x_{t}(J,q))-\ell_t(x_{t}(I,q))\right]\\\\
> > &\ne 0
> > \end{align*}
> > hence the argument for $\sum_{I\in S(t),q}w_t(I,q)r_t(I)\le 0$ seems flawed. So I'm still confused about what's  going on here.
> >
> > > we can remove the assumptions on and instead replace them by $G+1$ and $D+1$ in the bounds
> >
> > I'm not convinced that it is this simple, since now if $G<< 1$ the $(G+1)\sqrt{\log(T)}$ penalty can be significantly worse than
> > the proper $G\sqrt{\log(T)}$ penalty, so it seems either way one has either a suboptimal term or a non-standard assumption
> >
> > > this is because the optimal depends on , and to ensure we can find it using a binary search, the initial range should be set large enough which leads to within the logarithm
> >
> > I don't see why this is necessarily the case. $\eta^*$ always depends on G and D in these sorts of arguments yet still lead to $\log(T)$ experts instead of $\log(TGD)$ experts.
> > For instance, G and D don't turn up in the number of experts of [3], so I'm still skeptical that this isn't a bug of some sort
> >
> > > We address the experiment concerns as follows
> >
> > I see, thank you for the clarifications

---

> > > ### Author Response · Authors · 2022-11-16
> > > **Further response**
> > >
> > > Thank you for your detailed response!
> > >
> > > Regarding experts duplicates: $A_{I,i}$ is the same for different $i$ but varies for different $I$, therefore the regret bound is okay.  Indeed we only need one Adagrad instance for one interval $I$, then we treat them as many copies in the meta MW algorithm.
> > >
> > > Regarding the inequality: it is correct, since  $p_t(I,q)=w_t(I,q)/W_t$, splitting the two terms in RHS gives us $\sum_{J,q} w_t(J,q) \ell_t(x_t(J,q))-W_t \sum_{I,q} p_t(I,q) \ell_t(x_t(I,q))=0$. We will add this to make the proof more clear.
> > >
> > > Regarding the $G,D$ dependence: we agree some assumptions on $G,D$ can be lifted, leading to some minor improvements.  Specifically, the number of experts $Q$ doesn't need to have $G,D$ in the logarithm, and as a result the assumption that $G,D \ge 1$ can be lifted. In the middle of page 13 we have an expression for the optimal $\eta$, and we want to ensure $Q$ is large enough to cover it: $1/2GD 2^Q\le 4\sqrt{\log(T)/dTG^2D^2}$, which leads to $Q=O(\log (dT))$ (we missed the $GD$ term in the denominator of $\eta_{I,q}$), then we no longer need to assume $G,D\ge 1$ to make the logarithm non-negative. The $G^2D^2$ terms in the logarithm on page 12 can be removed as well. We thank you for pointing these improvements, we will add them.

---

> > > > ### Comment · Reviewer_fGkK · 2022-11-19
> > > > **Further response**
> > > >
> > > > Ah whoops, yes I see it now. With this and with the G>1 and D>1 and the GD factors removed from the logarithm I am much more confident about the correctness of the analysis.
> > > >
> > > > As stated before the main contribution is [3], but I haven't seen the specific trick used here before and it's amusing that it works.
> > > >
> > > > All things considered I have  raised my score considerably. The meta algorithm introduces a simple/neat trick, which I'm always a fan of, but overall I think the conflict with [3] places a lot of emphasis on the experimental results for novel contributions. I do think the experiments serve as a reasonable sanity check for the proposed algorithm, but to me they don't seem to constitute a novel contribution in their own right

---

### Official Review · Reviewer_dUbb · 2022-10-24

**Confidence:** 4
**Clarity, Quality, Novelty And Reproducibility:** No concerns here
**Correctness:** 4
**Technical Novelty And Significance:** 1
**Empirical Novelty And Significance:** 2
**Recommendation:** 3

**Strength And Weaknesses:**


So far as I can tell, the results appear to be correct and are clearly presented and well-motivated. However, I have some concerns about the quality of the theoretical bounds.

In particular, the main term in the regret bound of $\sqrt{d} \inf_{H\in\mathcal{H}} \sqrt{\sum_{t=a}^b g_t^\top H^{-1} g_t}$ is equal to $\text{trace}\left(\sqrt{\sum_{t=a}^b g_tg_t^\top}\right)$ (by https://jmlr.org/papers/volume12/duchi11a/duchi11a.pdf lemma 15).

Now, by applying the identity $\sum_{i} \sqrt{\lambda_i} \ge \sqrt{\sum_i \lambda_i}$ to the eigenvalues of $\sum_{t=a}^b g_tg_t^\top$, we have: $$\text{trace}\left(\sqrt{\sum_{t=a}^b g_tg_t^\top}\right)\ge \sqrt{\text{trace}\left(\sum_{t=a}^b g_tg_t^\top\right)}=\sqrt{\sum_{t=a}^b \|g_t\|^2}$$ where $\|\cdot\|$ indicates the 2-norm. Thus, the proposed bound of $D \sqrt{d} \inf_{H\in \mathcal{H}}\sqrt{\sum_{t=a}^b g_t^\top H^{-1} g_t}$ appears to never improve upon $D\sqrt{\sum_{t=a}^b \|g_t\|^2}$.  However, this bound is relatively easily obtainable without resorting to expensive matrix operations (e.g. in Cutkosky 2019 as cited in the paper. I believe also the analysis of Zhang et al 2018 https://arxiv.org/pdf/1906.10851.pdf can be tweaked to achieve this).

See for example http://blog.wouterkoolen.info/GrokkingAdaGrad/post.html for more discussion of this issue. So, while I am willing to believe that there is a possible improvement in the regret here, I do not see how this analysis brings it out. Instead, the theoretical result appears to be a somewhat worse regret bound with a significantly less efficient algorithm.

Some possibilities in this regard:

While the Duchi et al. 2011 does propose an example in which full-matrix adagrad performs better than a scalar learning rate, this analysis involved a special-case analysis of the dynamics of the algorithm rather than simply comparing regret bounds. Perhaps there is some way to modify this analysis to derive a general regret expression that can be applied here.

Alternatively, Cutkosky 2020 (https://proceedings.neurips.cc/paper/2020/hash/6495cf7ca745a9443508b86951b8e33a-Abstract.html) suggests that one can improve the analysis of adagrad by tuning the scalar part of the learning rate more carefully. Given the focus on learning rate adaptation in the experiments, perhaps the analysis could be improved to show a similar or better result in theory.


Beyond this issue, there should ideally be some discussion of Zhang et al 2018 https://arxiv.org/pdf/1906.10851.pdf  which achieves a different full-matrix bound of: $$\sum_{t=1}^T \langle g_t, w_t - u\rangle \le \tilde O\left(\sqrt{d \sum_{t=a}^b \langle g_t, w_t-u\rangle^2}\right).$$ Since both this paper and that one attempt to decrease the meta regret beyond $O(\sqrt{T})$ to obtain a better bound, it would be valuable to describe the differences.


Regarding the experiments: I appreciate the promise here, but I do not feel that these are convincing enough to overcome the analytical issues. With particular regard to the offline deep learning experiments, I am concerned that the algorithm will not scale to larger models. My understanding is that for modern extremely large models, even the space requirements that are a small linear multiple of the model size (e.g. as in Adam) can become difficult to accommodate, so I would much prefer to see convincing evidence that the method can scale up well.


**Summary Of The Paper:**

This paper considers the classic online convex optimization game and proposes a method to obtain “full matrix” regret bounds in a “strongly adaptive” sense. That is, the algorithm outputs $w_t\in W\subset \mathbb{R}^d$ in response to vectors $g_1,\dots,g_{t-1}$ such that for any interval [a,b], the algorithm ensures:
$$\sup_{u\in W} \sum_{t=a}^b \langle g_t, w_t - u\rangle \le \tilde O(D \sqrt{d} \inf_{H\in \mathcal{H}}\sqrt{\sum_{t=a}^b g_t^\top H^{-1} g_t}$$

Where $\mathcal{H}$ is the set of symmetric PSD matrices of trace at most $d$ and $D$ is the L2 diameter of $W$.

Experiments are presented on problems with explicit drift and in offline deep learning settings.


**Summary Of The Review:**

The theoretical results appear to not show advantage over prior methods. Given the focus of much of the paper on theory, this seems a significant issue.

---

> ### Author Response · Authors · 2022-11-14
> **Response**
>
> We thank you for your detailed feedback! We agree that the full-matrix adagrad bound does not improve upon second-order sgd bound in general. However, notice that the diagonal adagrad bound can be better than the sgd bound, up to $\sqrt{d}$, depending on the geometry of domain and sparsity of gradients (see the discussion of corollary 1 in the original adagrad paper)
>
> This renders our theoretical result   meaningful and improves previous bounds in certain cases. Notice that this bound follows from our proof in the following way. First step: simply replace experts by diagonal adagrads.  Next, we conclude a diagonal adagrad  bound for the meta MW algorithm as follows. On page 13 we want to bound the $\|x_{\tau}-x_{\tau}(I)\|^2_H$ term, if we restrict $H$ to be the class of diagonal PSD matrices with $\text{Tr}(H)\le d$ instead, this term is bounded by $d \|x_{\tau}-x_{\tau}(I)\|^2_{\infty}\le d D_{\infty}^2$ which gives the desired diagonal adagrad like bound.
>
> Regarding Zhang et al 18: unfortunately we were not aware of this paper. We agree that this paper can be used to derive our main result as a corollary. Note that our focus is on using adaptive regret for learning the learning rate schedule, and we believe this idea along with the experiments still has (practical) merit.
>
> Regarding experiment scalability to large models in offline setting, each expert - optimizer state, can be assigned to its own accelerator devices and all experts can run in a parallel fashion. This expert parallelism paradigm is well studied and adopted in natural language processing literature such as switch transformer \url{https://www.jmlr.org/papers/volume23/21-0998/21-0998.pdf}, and branch-train-merge \url{https://arxiv.org/pdf/2208.03306.pdf}.

---

### Decision · Program_Chairs · 2023-01-20

**Decision:**

Reject

**Justification For Why Not Higher Score:**

There is a previous paper with a stronger result.

**Justification For Why Not Lower Score:**

N/A

**Metareview: Summary, Strengths And Weaknesses:**

The paper have been carefully reviewed by experts in the field and found to be incremental with respect to prior work. In particular, two reviewers pointed out a paper that allow to derive the main result of this submission as a corollary. Given that in the rebuttal the authors acknowledged that they didn't know this paper, I encourage the authors to take it into account in their next iteration of their work.